# Harnessing digital health to objectively assess cognitive impairment in people undergoing hemodialysis process: The Impact of cognitive impairment on mobility performance measured by wearables

He Zhou[1], Fadwa Al-Ali[2], Changhong Wang[1], Abdullah Hamad[2], Rania Ibrahim[2], Talal Talal[3], Bijan Najafi[1]*

1 Interdisciplinary Consortium on Advanced Motion Performance (iCAMP), Michael E. DeBakey Department of Surgery, Baylor College of Medicine, Houston, Texas, United States of America, 2 Fahad Bin Jassim Kidney Center, Department of Nephrology, Hamad General Hospital, Hamad Medical Corporation, Doha, Qatar, 3 Diabetic Foot and Wound Clinic, Hamad Medical Corporation, Doha, Qatar

* najafi.bijan@gmail.com

**Data Availability Statement:** The minimal data set is available from the Data Archiving and

## Abstract

Cognitive impairment is prevalent but still poorly diagnosed in hemodialysis adults, mainly because of the impracticality of current tools. This study examined whether remotely monitoring mobility performance can help identifying digital measures of cognitive impairment in hemodialysis patients. Sixty-nine diabetes mellitus hemodialysis patients (age = 64.1 ±8.1years, body mass index = 31.7±7.6kg/m$^2$) were recruited. According to the Mini-Mental State Exam, 44 (64%) were determined as cognitive-intact, and 25 (36%) as cognitive-impaired. Mobility performance, including cumulated posture duration (sitting, lying, standing, and walking), daily walking performance (step and unbroken walking bout), as well as postural-transition (daily number and average duration), were measured using a validated pendant-sensor for a continuous period of 24-hour during a non-dialysis day. Motor capacity was quantified by assessing standing balance and gait performance under single-task and dual-task conditions. No between-group difference was observed for the motor capacity. However, the mobility performance was different between groups. The cognitive-impaired group spent significantly higher percentage of time in sitting and lying (Cohens effect size $d$ = 0.78, $p$ = 0.005) but took significantly less daily steps ($d$ = 0.69, $p$ = 0.015) than the cognitive-intact group. The largest effect of reduction in number of postural-transition was observed in walk-to-sit transition ($d$ = 0.65, $p$ = 0.020). Regression models based on demographics, addition of daily walking performance, and addition of other mobility performance metrics, led to area-under-curves of 0.76, 0.78, and 0.93, respectively, for discriminating cognitive-impaired cases. This study suggests that mobility performance metrics could be served as potential digital biomarkers of cognitive impairment among hemodialysis patients. It also highlights the additional value of measuring cumulated posture duration and postural-transition to improve the detection of cognitive impairment. Future studies need to examine

Networking Services (DANS) public repository (DOI: https://doi.org/10.17026/dans-xy5-n8c8).

**Funding:** Support was provided by the Qatar National Research Foundation (Award numbers: NPRP 7-1595-3-405 and NPRP 10-0208-170400). There was no additional external funding received for this study. The content is solely the responsibility of the authors and does not necessarily represent the official views of the sponsor.

**Competing interests:** Author TT is an employee of the Hamad Medical Co. Doha, Qatar' and 'Author Bijan Najafi is a handling editor on the PLOS ONE Digital Health Technology Call for Papers'. This does not affect our adherence to PLOS ONE policies on sharing data and materials.

potential benefits of mobility performance metrics for early diagnosis of cognitive impairment/dementia and timely intervention.

## Introduction

With aging of population, the burden of cognitive impairment appears to increase among patients with end-stage renal disease (ESRD) undergoing hemodialysis (HD) [1, 2]. As more patients of older age receive HD, cognitive impairment has become highly prevalent in this population [3–6]. At the same time, HD-associated factors can also increase the risk of cognitive impairment and cognitive decline among HD patients [4, 5]. Cognitive impairment leads to overall diminished quality of life and high medical costs associated with coexisting medical conditions and expensive care [7]. Early detection and routine assessment of cognitive function become crucial for delaying further cognitive decline in HD patients [8].

Ideally, HD patients should undergo routine screenings of cognitive function. However, routine assessments using current tools, such as Mini-Mental State Exam, (MMSE) [9], Montreal Cognitive Assessment (MoCA) [10], and Trail Making Test (TMT) [11], need be administered in a clinical setting under the supervision of a well-trained professional. Studies have reported that the accuracy and reliability of these screening tools depend on the experience and skills of the examiner, as well as the individual's educational level [12, 13]. Usually, in a regular dialysis clinic, the nurse does not equip with the professional experience or skills. Regular referral to a neuropsychological clinic could be also impractical as many HD patients have limited mobility, suffer from post-dialysis fatigue, and rarely accept to go to different locations than their regular dialysis clinics for the purpose of cognitive screening. Thus, it is not surprising that emerging literature has demonstrated that although cognitive impairment commonly occurs in HD population, it is still poorly diagnosed [14, 15].

"Mobility performance" depicts enacted mobility in real-life situations [16]. It is different than "motor capacity", which refers to an individual's motor function assessed under supervised condition [16]. Mobility performance requires multifaceted coordination between different parts of neuropsychology [17]. This includes motor capacity, intimate knowledge of environment, and difficulty of navigation through changing environments [18]. Understanding the association between mobility performance and cognitive function could help to design an objective tool for remote and potentially early detection of cognitive decline. Previous studies have demonstrated that in older adults, people with cognitive impairment exhibit lower level of activity [19–21]. However, in previous studies, the assessment of mobility performance mainly relied on self-reported questionnaires [19–21], Actigraphy [22, 23], or accelerometer-derived step count [24]. Although self-reported questionnaire is easy to access without the need of any equipment or device, its main limitation is lacking of objectivity [25]. Previous studies using Actigraphy or step count only provided limited information about mobility performance (activity level and daily step). They also neglected information about posture and postural-transition, which have been demonstrated to be more reliable than activity level or number of daily steps [26]. Considering the motor capacity in patients undergoing HD is usually deteriorated [27], and these patients are highly sedentary with reduced daily activity level [27], it may not be efficient enough to capture cognitive impairment in HD population by just using activity level or step count alone.

In this study, we used a pendant-like wearable sensor to mine potential digital biomarkers from mobility performance for capturing cognitive impairment and tracking the cognitive

decline in HD population. We measured detailed metrics of mobility performance including cumulated posture duration (sitting, lying, standing, and walking), daily walking performance (step count and number of unbroken walking bout), as well as postural-transition (daily number and average duration). We hypothesized that 1) HD patients with cognitive impairment have lower mobility performance than those without cognitive impairment; 2) the mobility performance derived digital biomarkers can determine cognitive impairment in HD patients, yielding better results than using daily walking performance alone.

## Materials and methods

### Study population

This study is a secondary analysis of a clinical trial focused on examining the benefit of exercise in adult HD patients (ClinicalTrials.gov Identifier: NCT03076528). The clinical trial was offered to all eligible HD patients visited the Fahad Bin Jassim Kidney Center (Hamad Medical Corporation, Doha, Qatar) for HD process. To be eligible, the subject should be a senior (age 50 years or older), be diagnosed with diabetes and ESRD that require HD, and have capacity to consent. Subjects were excluded if they had major amputation; were non-ambulatory or had severe gait or balance problem (e.g., unable to walk a distance of 15-meter independently with or without assistive device or unable to stand still without moving feet), which may affect their daily physical activity; had active foot ulcer or active infection; had major foot deformity (e.g. Charcot neuroarthropathy); had changes in psychotropic or sleep medications in the past 6-week; were in any active intervention (e.g. exercise intervention); had any clinically significant medical or psychiatric condition; or were unwilling to participate. All subjects signed a written consent approved by the Institutional Review Board at the Hamad Medical Corporation in Doha, Qatar. For the final data analysis, we only included those who had at least 24-hour valid mobility performance data during a non-dialysis day. Only baseline data without any intervention was used for the purpose of this study.

### Demographics, clinical data, and motor capacity

Demographics and relevant clinical information for all subjects were collected using chart-review and self-report, including age, gender, height, weight, fall history, duration of HD, and daily number of prescription medicines. Body mass index (BMI) was calculated based on height and weight information.

All subjects underwent clinical assessments, including MMSE [9], Center for Epidemiologic Studies Depression scale (CES-D) [28], Physical Frailty Phenotype [29], neuropathy screening using Vibration Perception Threshold test (VPT) [30], vascular assessment using Ankle Brachial Index test (ABI) [31], and glycated hemoglobin test (HbA1c) [32]. The CES-D short-version scale was used to measure self-reported depression symptoms. A cutoff of CES-D score of 16 or greater was used to identify subjects at risk for clinical depression [28]. The Physical Frailty Phenotype, including unintentional weight loss, weakness (grip strength), slow gait speed (15-foot gait test), self-reported exhaustion, and self-reported low physical activity, was used to assess frailty [29]. Subjects with 1 or 2 positive criteria were considered pre-frail, and those with 3 or more positive criteria were considered frail. Subjects negative for all criteria were considered robust [29]. Plantar numbness was evaluated by the VPT measured on six plantar regions of interest, including the left and right great toes, 5th metatarsals, and heels. In this study, we used the maximum value of VPT measures under regions of interest for both feet to evaluate the Diabetic Peripheral Neuropathy (DPN) status. A subject was designated with DPN if his/her maximum VPT reached 25 volts or greater [30]. The ABI was calculated as the ratio of the systolic blood pressure measured at the ankle to the systolic blood pressure

measured at the upper arm. A subject was designated with the Peripheral Artery Disease (PAD) if his/her ABI value was either greater than 1.2 or smaller than 0.8 [31].

Motor capacity was quantified by assessing standing balance and walking performance [33]. Standing balance was measured using wearable sensors (LegSys[TM], BioSensics LLC., MA, USA) attached to lower back and dominant front lower shin. Subject stood in the upright position, keeping feet close together but not touching, with arms folded across the chest, for 30-second. Center of mass sway (unit: cm$^2$) was calculated using validated algorithms [34]. We assessed walking performance under both single-task and dual-task conditions to determine the impact of cognitive impairment on motor capacity. Walking performance was measured using the same wearable sensors attached to both front lower shins. Subjects were asked to walk with their habitual gait speed for 15-meter with no cognitive task (single-task condition). Then, they were asked to repeat the test while loudly counting backward from a random number (dual-task condition: motor task + working memory) [33]. Gait speeds under both conditions were calculated using validated algorithms [35].

## Determination of cognitive impairment

Cognitive impairment was defined as a MMSE score less than 28 as recommended by Tobias et al. and Damian et al. studies [36, 37]. In these studies, researchers have demonstrated that MMSE cutoff score of 28 yields the highest sensitivity and specificity to identify those with cognitive impairment compared to the commonly used lower cutoff scores.

## Sensor-derived monitoring of mobility performance

Mobility performance was characterized by 1) cumulated posture duration, including percentage of sitting, lying, standing, and walking postures of 24-hour; 2) daily walking performance, including step count and number of unbroken walking bout (an unbroken walking bout was defined as at least three consecutive steps within 5 seconds interval [38]); and 3) postural-transition, including total number of postural-transition such as sit-to-stand, stand-to-sit, walk-to-stand, stand-to-walk, walk-to-sit (direct transition from walking to sitting with standing pause less than 1 seconds [39]), and sit-to-walk (direct transition from sitting to walking with standing pause less than 1 seconds [39]), as well as average duration of postural-transition (time needed for rising from a chair or sitting on a chair [40]). Mobility performance was recorded for a continuous period of 24-hour using a validated pendant sensor (PAMSys[TM], BioSensics LLC., MA, USA, Fig 1) worn during a non-dialysis day. We selected a non-dialysis day because the data during a day of dialysis could be biased by the long period of sitting/lying during HD process and the post dialysis fatigue. The PAMSys[TM] sensor contains a 3-axis accelerometer (sampling frequency of 50 Hz) and built-in memory for recording long-term data. The description of methods to extract metrics of interest was described in details in our previous studies [38–42].

## Statistical analysis

All continuous data was presented as mean ± standard deviation. All categorical data was expressed as percentage. Analysis of variance (ANOVA) was used for between-group comparison of continuous demographics and clinical data, as well as mobility performance metrics. Analysis of Chi-square was used for comparison of categorical demographics and clinical data. Analysis of covariance (ANCOVA) was employed to compare differences between groups for motor capacity metrics and mobility performance metrics, with adjustment for age and BMI. A 2-sided $p<0.050$ was considered to be statistically significant. The effect size for discriminating between groups was estimated using Cohen's $d$ effect size and represented as $d$ [43]. The

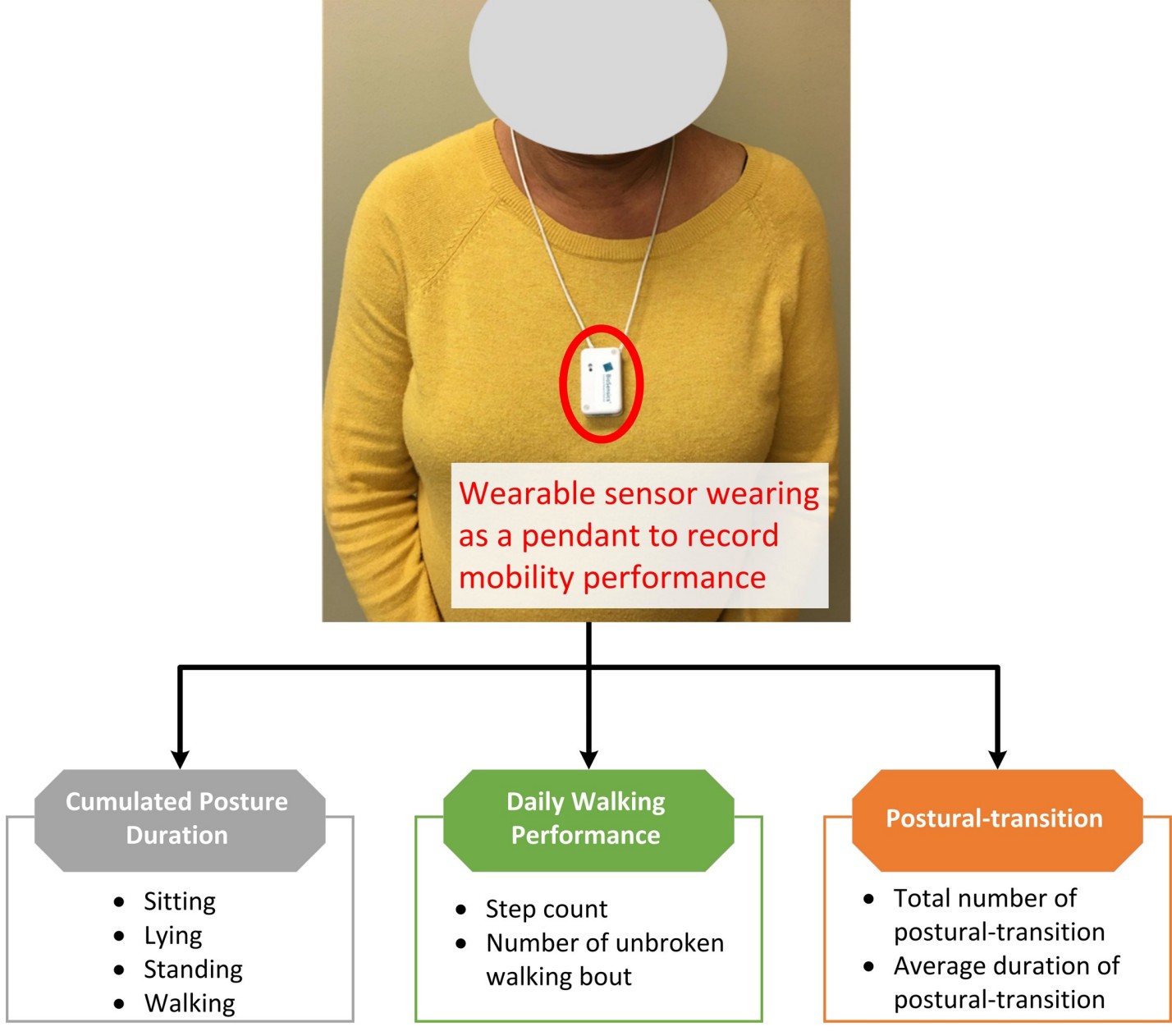

**Fig 1. A patient wearing the sensor as a pendant.** Detailed metrics of mobility performance, including cumulated posture duration (sitting, lying, standing, and walking), daily walking performance (step count and number of unbroken walking bout), as well as postural-transition (daily number and average duration), were measured.

Pearson correlation coefficient was used to evaluate the degree of agreement between mobility performance metrics and motor capacity variable for both groups with and without cognitive impairment. The correlation coefficient was also interpret as effect size [43, 44]. A multivariate linear regression model was used to determine the association between mobility performance metrics and MMSE. In this model, MMSE was the dependent variable, and mobility performance metrics and demographics were the independent variables. $R^2$ and $p$-value were calculated for the multivariate linear regression model. The Pearson correlation coefficient was

used to evaluate the degree of agreement between the regression model and MMSE. Further, binary logistic regression analysis was employed to examine the relationship between each study variable and cognitive impairment. First, univariate logistic regression was employed to investigate the relationship of the test variables using "cognitive-impaired/cognitive-intact" as the dependent variable. Nagelkerke R Square ($R^2$), odds ratio (OR), 95% confidence interval (95% CI), and $p$-value were calculated for each explanatory variable. Second, stepwise multivariate logistic regression, using variables found with $p<0.20$ in the univariate analysis, was performed to investigate independent effects of variables in predicting cognitive impairment. Then, these variables with independent effects were used to build models for prospective cognitive impairment prediction. In Model 1 (reference model), we only used demographics as independent variables. Then, to examine additional values of mobility performance metrics, two other models were examined. In Model 2, independent variables included demographics and daily walking performance. In Model 3, we added cumulated posture duration and postural-transition as additional independent variables. The receiver operating characteristic (ROC) curve and area-under-curve (AUC) were calculated for prediction models. All statistical analyses were performed using IBM SPSS Statistics 25 (IBM, IL, USA).

## Results

Eighty-one subjects satisfied the inclusion and exclusion criteria of this study. However, the mobility performance data was available and valid for 69 subjects. Reasons of unavailable and invalid mobility performance data were refusal of wearing the sensor (n = 9) and wearing duration less than 24-hour (n = 3). Table 1 summarizes demographics, clinical data, and motor capacity of the remaining subjects. According to the MMSE, 44 subjects (64%) were classified as cognitive-intact, and 25 (36%) were classified as cognitive-impaired. The average MMSE score of the cognitive-impaired group was 22.6±3.7, which was significantly lower than the cognitive-intact group with 29.2±0.9 ($p<0.001$). The cognitive-impaired group was significantly older than the cognitive-intact group ($p = 0.001$). Female percentage was significantly higher in the cognitive-impaired group ($p = 0.008$). The cognitive-impaired group was shorter than the cognitive-intact group ($p = 0.009$). But there was no between-group difference regarding the BMI. No between-group difference was observed for subjects' weight, fall history, duration of HD, number of prescription medications, prevalence of at risk for clinical depression, prevalence of frailty and pre-frailty, VPT, prevalence of DPN, prevalence of PAD, and HbA1c ($p>0.050$). No between group difference was observed for motor capacity metrics including standing balance and walking performance ($p>0.050$). For the dual-task walking, the cognitive-impaired group had lower dual-task walking speed than the cognitive-intact group. But the difference did not reach statistical significance.

Table 2 summarizes between-group comparison for mobility performance metrics during 24-hour. The cognitive-impaired group spent significantly higher percentage of time in sitting and lying ($d = 0.78$, $p = 0.005$, Fig 2) but spent significantly lower percentage of time in standing ($d = 0.70$, $p = 0.010$, Fig 2) and walking ($d = 0.77$, $p = 0.007$, Fig 2). They also took significantly less steps ($d = 0.69$, $p = 0.015$) and unbroken walking bout ($d = 0.56$, $p = 0.048$) than the cognitive-intact group. Longer durations of sit-to-stand transition ($d = 0.37$, $p = 0.143$) and stand-to-sit transition ($d = 0.50$, $p = 0.044$) were observed in the cognitive-impaired group. Significant reductions of number of postural-transition were also observed in the cognitive-impaired group, including total number of transition to walk ($d = 0.60$, $p = 0.035$), number of stand-to-walk transition ($d = 0.60$, $p = 0.036$), number of walk-to-sit transition ($d = 0.65$, $p = 0.020$), total number of transition to stand ($d = 0.62$, $p = 0.024$), and number of walk-to-stand transition ($d = 0.58$, $p = 0.044$). When results were adjusted by demographic covariates

**Table 1. Demographics, clinical data, and motor capacity of the study population.**

| | Cognitive-Intact (n = 44) | Cognitive-Impaired (n = 25) | *p-value* |
|---|---|---|---|
| **Demographics** | | | |
| Age, *years* | 61.8 ± 6.7 | 68.1 ± 8.8 | 0.001* |
| Sex (Female), *%* | 43% | 76% | 0.008* |
| Height, *m* | 1.63 ± 0.09 | 1.50 ± 0.29 | 0.009* |
| Weight, *kg* | 83.4 ± 21.5 | 76.3 ± 16.6 | 0.156 |
| Body Mass Index, *kg/m²* | 31.8 ± 8.6 | 31.4 ± 5.4 | 0.804 |
| **Clinical data** | | | |
| Had fall in last 12-month, *%* | 21% | 36% | 0.158 |
| Duration of HD, *years* | 4.6 ± 5.4 | 3.5 ± 2.3 | 0.354 |
| Number of prescription medications, *n* | 8 ± 3 | 8 ± 3 | 0.233 |
| Mini-mental State Exam, *units on a scale* | 29.2 ± 0.9 | 22.6 ± 3.7 | <0.001* |
| Center for Epidemiologic Studies Depression, *units on a scale* | 13.1 ± 6.3 | 16.0 ± 12.6 | 0.209 |
| At risk for clinical depression, *%* | 27% | 44% | 0.157 |
| Robust, *%* | 2% | 0 | 0.448 |
| Pre-frailty & frailty, *%* | 98% | 100% | 0.448 |
| Vibration Perception Threshold, *V* | 32.1 ± 16.5 | 34.6 ± 16.0 | 0.544 |
| Diabetic Peripheral Neuropathy, *%* | 61% | 68% | 0.534 |
| Peripheral Arterial Disease, *%* | 56% | 68% | 0.322 |
| Glycated Hemoglobin, *%* | 6.7 ± 1.5 | 6.6 ± 1.3 | 0.783 |
| **Motor Capacity †** | | | |
| Static balance (center of mass sway), *cm²* | 0.39 ± 0.38 | 0.21 ± 0.39 | 0.087 |
| Single-task walking speed, *m/s* | 0.49 ± 0.19 | 0.44 ± 0.20 | 0.345 |
| Dual-task walking speed, *m/s* | 0.46 ± 0.19 | 0.43 ± 0.19 | 0.682 |

At risk for clinical depression was assessed by Center for Epidemiologic Studies Depression score with a cutoff of 16 or greater

Diabetic Peripheral Neuropathy was assessed by maximum Vibration Perception Threshold value with a cutoff of 25-volt or greater

*: significant difference between groups

†: Results were adjusted by age and BMI

including age and BMI, several mobility performance metrics remained significant for comparing between the cognitive-impaired and cognitive-intact groups (Table 2).

Fig 3 illustrates the correlation between motor capacity and mobility performance among HD patients with and without cognitive impairment. A significant correlation with medium effect size was observed between single-task walking speed and number of stand-to-sit transition among HD patients without cognitive impairment ($r = 0.39$, $p = 0.012$, Fig 3A). But the correlation among cognitive-impaired subjects was insignificant ($r = -0.18$, $p = 0.417$). Similarly, a significant correlation with medium effect size was observed between single-task walking speed and number of sit-to-stand transition among HD patients without cognitive impairment ($r = 0.42$, $p = 0.006$, Fig 3B). But the correlation was diminished among cognitive-impaired subjects ($r = -0.19$, $p = 0.378$).

Results from the multivariate linear regression model ($R^2 = 0.400$, $p = 0.019$) revealed that "age" (B = -0.225, $p<0.001$) and "average duration of sit-to-stand transition" (B = -4.768, $p = 0.017$) were independent predictors of MMSE. A significant correlation with large effect size of $r = 0.64$ ($p<0.001$) was determined between the regression model and MMSE (Fig 4).

In the univariate regression analysis, 5 variables in demographics and all variables in the mobility performance were associated with cognitive impairment ($p<0.20$) (Table 3). Two demographic variables and 11 mobility performance variables remained in the multivariate

**Table 2. Mobility performance (in 24-hour) comparison for cognitive-intact and cognitive-impaired groups.**

|  | Cognitive- Intact | Cognitive- Impaired | Mean Difference % | Cohen's d | p-value | Adjusted p-value † |
|---|---|---|---|---|---|---|
| **Cumulated Posture Duration** |  |  |  |  |  |  |
| Sitting + lying percentage, % | 82.0 ± 11.3 | 89.1 ± 6.3 | 9% | 0.78 | 0.005* | 0.028* |
| Standing percentage, % | 15.3 ± 9.2 | 9.9 ± 5.9 | -35% | 0.70 | 0.010* | 0.061 |
| Walking percentage, % | 2.6 ± 3.0 | 0.9 ± 0.9 | -65% | 0.77 | 0.007* | 0.010* |
| **Daily Walking Performance** |  |  |  |  |  |  |
| Step count, n | 1827 ± 2382 | 608 ± 688 | -67% | 0.69 | 0.015* | 0.024* |
| Number of unbroken walking bout, n | 62 ± 85 | 27 ± 25 | -57% | 0.56 | 0.048* | 0.083 |
| **Postural-transition** |  |  |  |  |  |  |
| Average duration of stand-to-sit transition, s | 2.9 ± 0.2 | 3.0 ± 0.2 | 3% | 0.37 | 0.143 | 0.128 |
| Average duration of sit-to-stand transition, s | 3.0 ± 0.2 | 3.1 ± 0.3 | 4% | 0.50 | 0.044* | 0.023* |
| Total number of transition to walk, n | 63 ± 89 | 24 ± 23 | -63% | 0.60 | 0.035* | 0.068 |
| Number of sit-to-walk transition, n | 8 ± 8 | 4 ± 5 | -44% | 0.51 | 0.061 | 0.183 |
| Number of stand-to-walk transition, n | 54 ± 82 | 19 ± 19 | -66% | 0.60 | 0.036* | 0.064 |
| Total number of transition to sit, n | 149 ± 71 | 119 ± 56 | -20% | 0.46 | 0.077 | 0.300 |
| Number of walk-to-sit transition, n | 13 ± 14 | 6 ±7 | -53% | 0.65 | 0.020* | 0.039* |
| Number of stand-to-sit transition, n | 108 ± 64 | 88 ± 51 | -18% | 0.34 | 0.186 | 0.561 |
| Total number of transition to stand, n | 175 ± 107 | 121 ± 61 | -31% | 0.62 | 0.024* | 0.094 |
| Number of sit-to-stand transition, n | 111 ± 68 | 87 ± 50 | -22% | 0.40 | 0.126 | 0.456 |
| Number of walk-to-stand transition, n | 50 ± 78 | 17 ± 17 | -65% | 0.58 | 0.044* | 0.083 |

Effect sizes were calculated as Cohen's d

*: significant difference between groups

†: Results were adjusted by age and BMI

model suggesting that they are independent predictors (Table 3). These variables were used to build regression models. ROC curves for the 3 models were displayed in Fig 5. The AUC for

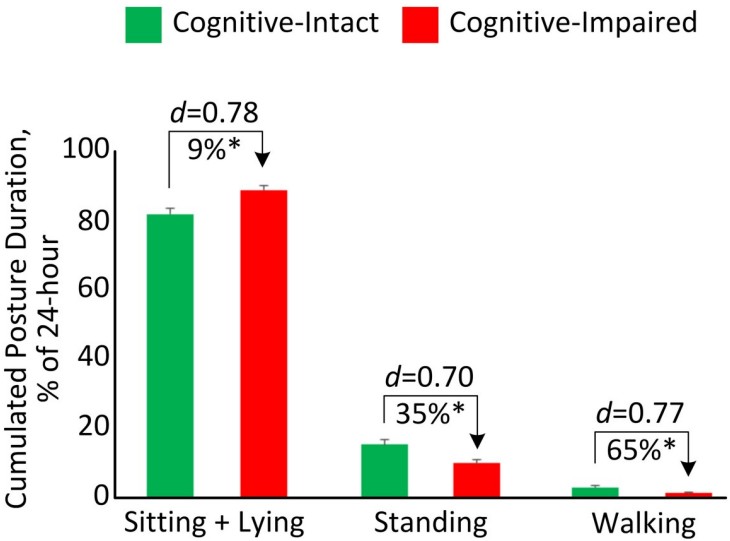

**Fig 2. Cumulated posture duration (as percentage of 24-hour) for the cognitive-intact group and cognitive-impaired group.** Error bar represents the standard error. "*d*" denotes the Cohen's *d* effect size. "*" denotes when the between-group comparison achieved a statistically significant level ($p < 0.050$).

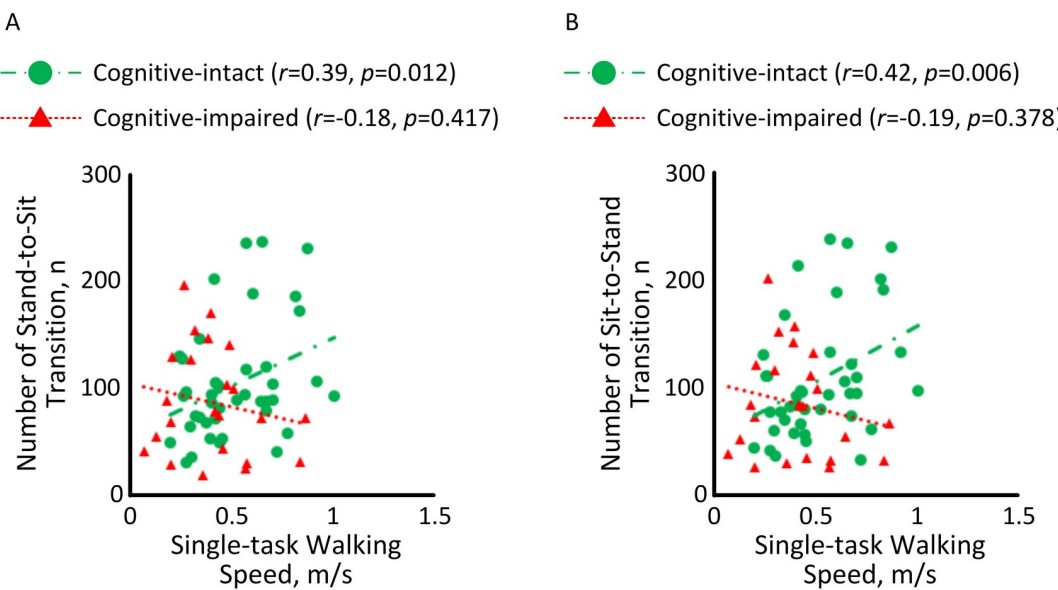

**Fig 3.** Correlations between single-task walking speed and (A) number of stand-to-sit transition and (B) number of sit-to-stand transition among HD patients with and without cognitive impairment.

Model 1 (demographics alone) was 0.76, with a sensitivity of 44.0%, specificity of 88.6%, and accuracy of 72.5% for predicting cognitive impairment. The AUC for Model 2 (demographics + daily walking performance) was 0.78, with a sensitivity of 44.0%, specificity of 79.5%, and

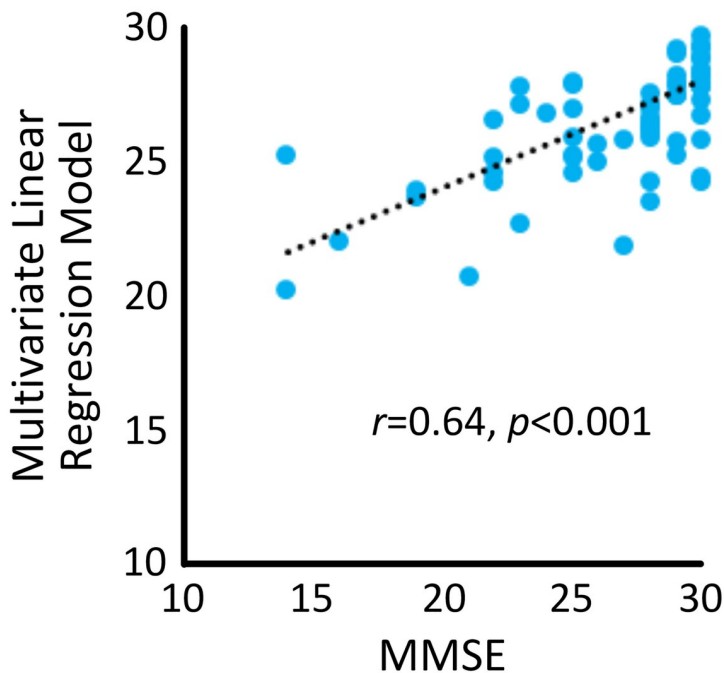

**Fig 4. A significant correlation was observed between the multivariate linear regression model and MMSE.**

**Table 3. Results of univariate and multivariate logistic regression.**

| | $R^2$ | OR | 95% CI | *p-value* |
|---|---|---|---|---|
| **Demographics** | | | | |
| Age | 0.190 | 1.116 | 1.036–1.201 | 0.004^ |
| Sex | 0.136 | 4.167 | 1.394–12.451 | 0.011 |
| Height | 0.206 | 0.917 | 0.862–0.975 | 0.006^ |
| Weight | 0.044 | 0.980 | 0.952–1.008 | 0.161 |
| BMI | 0.001 | 0.992 | 0.928–1.059 | 0.800 |
| Had fall in last 12-month | 0.038 | 2.187 | 0.730–6.552 | 0.162 |
| Duration of HD | 0.017 | 0.940 | 0.816–1.084 | 0.396 |
| Number of prescription medications | 0.031 | 1.116 | 0.931–1.336 | 0.235 |
| **Cumulated Posture Duration** | | | | |
| Sitting + lying percentage | 0.167 | 1.094 | 1.022–1.172 | 0.010^ |
| Standing percentage | 0.141 | 0.907 | 0.838–0.982 | 0.016^ |
| Walking percentage | 0.174 | 0.642 | 0.441–0.935 | 0.021^ |
| **Daily Walking Performance** | | | | |
| Step count | 0.158 | 0.999 | 0.999–1.000 | 0.027 |
| Number of unbroken walking bout | 0.110 | 0.986 | 0.971–1.001 | 0.066^ |
| **Postural-transition** | | | | |
| Average duration of stand-to-sit transition | 0.042 | 4.515 | 0.583–34.965 | 0.149 |
| Average duration of sit-to-stand transition | 0.078 | 7.427 | 0.975–56.590 | 0.053^ |
| Total number of transitions to walk | 0.132 | 0.984 | 0.968–1.000 | 0.050^ |
| Number of sit-to-walk transition | 0.078 | 0.921 | 0.841–1.008 | 0.075 |
| Number of stand-to-walk transition | 0.136 | 0.981 | 0.963–1.000 | 0.051^ |
| Total number of transitions to sit | 0.068 | 0.992 | 0.983–1.001 | 0.083^ |
| Number of walk-to-sit transition | 0.121 | 0.935 | 0.880–0.994 | 0.032^ |
| Number of stand-to-sit transition | 0.038 | 0.994 | 0.984–1.003 | 0.190 |
| Total number of transitions to stand | 0.111 | 0.993 | 0.986–0.999 | 0.031 |
| Number of sit-to-stand transition | 0.051 | 0.993 | 0.983–1.002 | 0.133^ |
| Number of walk-to-stand transition | 0.130 | 0.979 | 0.959–1.001 | 0.056^ |

^: Variables remained in the multivariate model

accuracy of 66.7% for predicting cognitive impairment. The highest AUC (0.93) was obtained by Model 3 (demographics + daily walking performance + cumulated posture duration + postural-transition), with a sensitivity of 72.0%, specificity of 93.2%, and accuracy of 85.5% for distinguishing cognitive-impaired cases.

## Discussions

To our knowledge, this is the first study to investigate the association between mobility performance and cognitive condition in patients with diabetes and ESRD undergoing HD process. The results suggest that although HD patients with and without cognitive impairment have similar motor capacity, those with cognitive impairment have lower mobility performance. We were able to confirm our hypothesis that mobility performance metrics during a non-dialysis day could be used as potential digital biomarkers of cognitive impairment among HD patients. Specifically, several mobility performance metrics measurable using a pendant sensor enable significant discrimination between those with and without cognitive impairment with medium effect size (maximum Cohen's $d$ = 0.78). In addition, a metric constructed by the combination of demographics and mobility performance metrics yields a significant

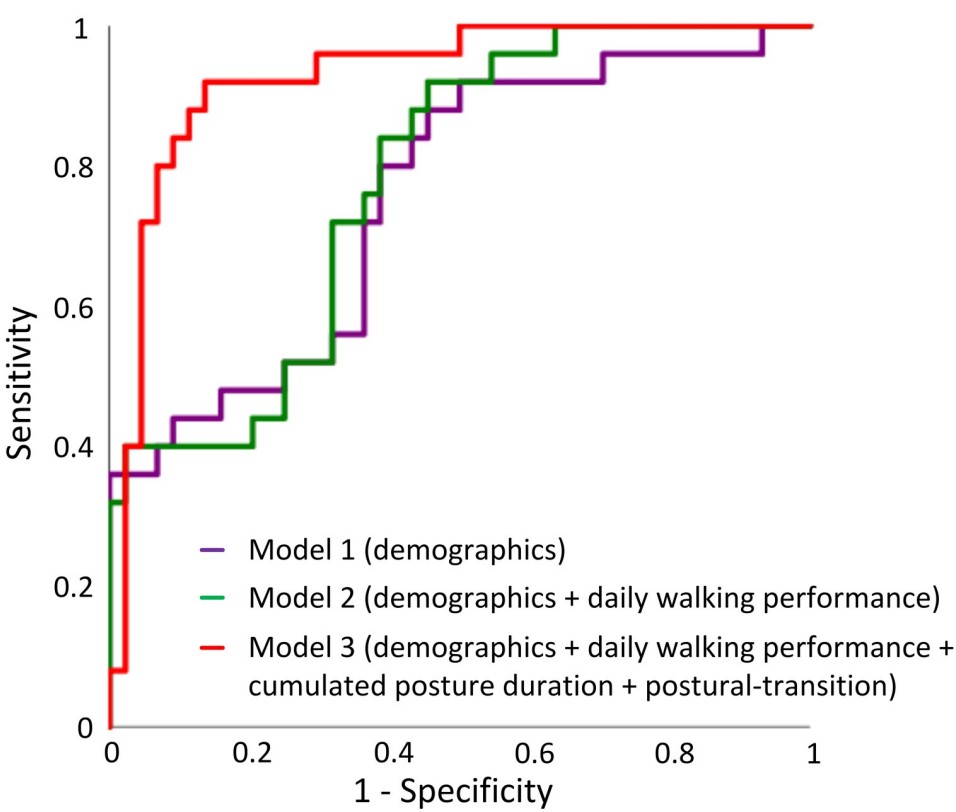

**Fig 5. ROCs of different models for predicting cognitive impairment: Model 1 used "demographics" (AUC = 0.76), Model 2 used a combination of "demographics" and "daily walking performance" (AUC = 0.78), and Model 3 used a combination of "demographics", "daily walking performance", "cumulate posture duration", and "postural-transition" (AUC = 0.93).**

correlation with large effect size with the MMSE ($r = 0.64$, $p < 0.001$). By adding mobility performance together with demographics into the binary logistic regression model, it enables distinguishing between those with and without cognitive impairment. This combined model yields relatively high sensitivity, specificity, and accuracy, which is superior to using demographics alone. Our results also suggest that despite cognitive-impaired HD patients have poor daily walking performance, just monitoring daily walking performance may not be sufficient to distinguish those with cognitive impairment. Additional mobility performance metrics, including cumulated posture duration and postural-transition, could increase the AUC from 0.78 to 0.93 for detection of cognitive-impaired cases.

Previous studies investigating association between mobility performance and cognitive impairment showed that activity level and daily steps are positively associated with cognitive function in older adults [19–24]. Results of this study are in line with the previous studies. They showed that cognitive-impaired HD patients have lower walking percentage and step count than cognitive-intact HD patients. Additionally, we found the cognitive-impaired HD patients have less number of postural-transition than cognitive-intact HD patients during daily living. The limited number of postural-transition has been identified as a factor which may contribute to the muscle weakness and activity limitations, causing physical frailty [39, 45]. Frailty together with cognitive impairment (known as 'cognitive frailty') has been shown to be a strong and independent predictor of further cognitive decline over time [46, 47].

Mobility performance in daily life depends not only on motor capacity, but also on intact cognitive function and psychosocial factors [48]. Studies have shown that cognitive impairment is associated with reduced mobility performance [48–50]. However, an individual's scores in supervised tests are poorly related to mobility performance in real life [48–50]. Results of this study show that among cognitive-intact HD patients, mobility performance is associated with motor capacity. However among HD patients with cognitive impairment, motor capacity is poorly related to mobility performance. This demonstrates that cognitive function is a moderator between motor capacity and mobility performance among patients undergoing HD process. This is aligned with the study of Feld et al. [51], in which it was demonstrated that gait speed does not adequately predict whether stroke survivors would be active in the community. Similar observation was reported by Toosizadeh et al. study [52], in which no agreement between motor capacity and mobility performance was observed among people with Parkinson's disease, while a significant agreement was observed among age-matched healthy controls.

In previous studies, to better link motor capacity with cognitive decline, dual-task walking test was proposed [53]. By adding cognitive challenges into motor task, the dual-task walking speed can expose cognitive deficits through the evaluation of locomotion. Previous studies have shown that dual-task walking speed for cognitive-impaired older adults was statistically lower than cognitive-intact ones among non-dialysis population [54]. Surprisingly, we didn't observe significant between-group difference in our sample. A previous systematic review has pointed out that older adults with mobility limitation are more likely to prioritize motor performance over cognitive performance [55]. We speculate that because of the poor motor capacity among HD population, subjects would prioritize motor task over cognitive task. Thus the effect of cognitive impairment may not be noticeable in this motor-impaired population by dual-task walking speed. If this can be confirmed in the follow up study, it may suggest that dual-task paradigm may not be a sufficient test to determine cognitive deficit among population with poor motor capacity.

In this study, we found the cognitive-impaired group had higher percentage of female. This finding is in line with the previous studies [56, 57]. For example, Beam et al. examined gender differences in incidence rates of any dementia, Alzheimer's disease (AD) alone, and non-Alzheimer's dementia alone in 16926 women and men in the Swedish Twin Registry aged 65+. They reported that incidence rates of any dementia and AD were greater in women than men, particularly in older ages (age of 80 years and older) [56]. Similarly, Wang et al. suggested that females compared to males showed significantly worse performance in cognitive function [57]. In this study, we did not adjust the results by gender because previous studies have demonstrated that gender does not affect mobility performance in HD population [58–61].

A major limitation of this study is the relatively low sample size, which could be underpowered for the clinical conclusion. On the other hand, this study could be considered as a cohort study as all participants were recruited from the Fahad Bin Jassim Kidney Center of Hamad Medical Corporation, which supports the majority of HD patients in the state of Qatar. All eligible subjects who received HD in this center were offered to participate in this study. Another limitation of this study is that mobility performance metrics were only measured in a single non-dialysis day. We excluded mobility performance monitoring during the dialysis day because we anticipated that data could be biased by the long process of HD (often 4-hour). Patients are holding a sitting or lying posture during the HD process. They also suffer the post-dialysis fatigue on the dialysis day. In addition, the measured single-day mobility performance may not be able to accurately represent the condition of HD patients (including both weekdays and weekends). Several previous literature reported three or more days of accelerometry data may more reliably and accurately model mobility performance in adult population [62, 63]. It

would be interesting to investigate whether multiple days of monitoring could model mobility performance more accurately in HD patients in the future study, since HD patients may have fluctuation in mobility performance due to post-dialysis fatigue and change of renal function [64].

## Conclusion

This study suggests that mobility performance metrics remotely measurable using a pendant sensor during a non-dialysis day could be served as potential digital biomarkers of cognitive impairment among HD patients. Interestingly, motor capacity metrics, even assessed under the cognitively demanding condition, are not sensitive to cognitive impairment among HD patients. Results suggest that despite cognitive-impaired HD patients have poor daily walking performance, just monitoring daily walking performance may not be sufficient to determine cognitive impairment cases. Additional mobility performance metrics such as cumulated posture duration and postural-transition can improve the discriminating power. Further researches are encouraged to evaluate the ability of sensor-derived mobility performance metrics to determine early cognitive impairment or dementia, as well as to track potential change in cognitive impairment over time in response to HD process. Future studies are also recommended for the potential use of sensor-derived metrics to determine modifiable factors, which may contribute in cognitive decline among HD patients.

## Acknowledgments

We thank Mincy Mathew, Priya Helena Peterson, Ana Enriquez, and Mona Amirmazaheri for assisting with data collection.

## Author Contributions

**Conceptualization:** Fadwa Al-Ali, Bijan Najafi.

**Data curation:** Abdullah Hamad, Rania Ibrahim, Talal Talal.

**Formal analysis:** He Zhou.

**Funding acquisition:** Fadwa Al-Ali, Bijan Najafi.

**Supervision:** Fadwa Al-Ali, Bijan Najafi.

**Writing – original draft:** He Zhou, Changhong Wang, Bijan Najafi.

**Writing – review & editing:** He Zhou, Fadwa Al-Ali, Changhong Wang, Abdullah Hamad, Rania Ibrahim, Talal Talal, Bijan Najafi.

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
