## [Decision Letter · Decision Letter 0]

16 Mar 2020

PONE-D-19-30554

Harnessing Digital Health to Objectively Assess Cognitive Impairment in People undergoing Hemodialysis Process: The Impact of Cognitive Impairment on Mobility Performance Measured by Wearables

PL

Thank you for submitting your manuscript to PLOS ONE. After careful consideration, we feel that it has merit but does not fully meet PLOS ONE’s publication criteria as it currently stands. Therefore, we invite you to submit a revised version of the manuscript that addresses the points raised during the review process.

We would appreciate receiving your revised manuscript by Apr 30 2020 11:59PM. To enhance the reproducibility of your results, we recommend that if applicable you deposit your laboratory protocols in protocols.io, where a protocol can be assigned its own identifier (DOI) such that it can be cited independently in the future. For instructions see: http://journals.plos.org/plosone/s/submission-guidelines#loc-laboratory-protocols

We look forward to receiving your revised manuscript.

Kind regards,

Luigi Lavorgna

Academic Editor

PLOS ONE

Journal Requirements:

"Partial support was provided by the Qatar National Research Foundation (Award number: NPRP 7-1595-3-405 and NPRP 10-0208-170400).".

i) Please provide an amended statement that declares *all* the funding or sources of support (whether external or internal to your organization) received during this study, as detailed online in our guide for authors at http://journals.plos.org/plosone/s/submit-now.  Please also include the statement “There was no additional external funding received for this study.” in your updated Funding Statement.

ii) Please include your amended Funding Statement within your cover letter. We will change the online submission form on your behalf.

"None".

We note that one or more of the authors are employed by a commercial company: 'Hamad Medical Corporation'.

Reviewers' comments:

Reviewer's Responses to Questions

**Comments to the Author**

1. Is the manuscript technically sound, and do the data support the conclusions?

Reviewer #1: Yes

2. Has the statistical analysis been performed appropriately and rigorously? 

Reviewer #1: Yes

3. Have the authors made all data underlying the findings in their manuscript fully available?

Reviewer #1: Yes

4. Is the manuscript presented in an intelligible fashion and written in standard English?

Reviewer #1: Yes

5. Review Comments to the Author

Reviewer #1: The study reported in this manuscript is a well-done clinical trial, the conduction of which has followed all the recommended steps by checklists and guidelines, including the registration of its protocol. The statistical analyses are robust and scientifically sound. I would only recommend authors to streamline introduction, that, as it is now, reads quite long.

6. PLOS authors have the option to publish the peer review history of their article (what does this mean?). If published, this will include your full peer review and any attached files.

Reviewer #1: No

---

## [Author Response · Author response to Decision Letter 0]

31 Mar 2020

Journal Requirements

C1. When submitting your revision, we need you to address these additional requirements.

http://www.journals.plos.org/plosone/s/file?id=wjVg/PLOSOne_formatting_sample_main_body.pdf and http://www.journals.plos.org/plosone/s/file?id=ba62/PLOSOne_formatting_sample_title_authors_affiliations.pdf. 

Response: Done

C2. In your Data Availability statement, you have not specified where the minimal data set underlying the results described in your manuscript can be found. PLOS defines a study's minimal data set as the underlying data used to reach the conclusions drawn in the manuscript and any additional data required to replicate the reported study findings in their entirety. All PLOS journals require that the minimal data set be made fully available. For more information about our data policy, please see http://journals.plos.org/plosone/s/data-availability.

Response: The de-identified data was uploaded in a public repository: Data Archiving and Networking Services (DANS). https://doi.org/10.17026/dans-xy5-n8c8

Relevant DOI has been added into the revised cover letter.

C3. Please provide an amended statement that declares *all* the funding or sources of support (whether external or internal to your organization) received during this study, as detailed online in our guide for authors at http://journals.plos.org/plosone/s/submit-now. Please also include the statement “There was no additional external funding received for this study.” in your updated Funding Statement.

Response: Done

Page 22 Line 435: “Support was provided by the Qatar National Research Foundation (Award number: NPRP 7-1595-3-405 and NPRP 10-0208-170400). There was no additional external funding received for this study. The content is solely the responsibility of the authors and does not necessarily represent the official views of the sponsor.”

C4. Please include your amended Funding Statement within your cover letter. We will change the online submission form on your behalf.

Response: Done.

C5. We note that one or more of the authors are employed by a commercial company: 'Hamad Medical Corporation'.

Response: Please note that Hamad Medical Corporation (HMC) is a non-profit public healthcare provider in the State of Qatar (https://www.hamad.qa/EN). None of the co-authors, including authors affiliated with HMC, claimed any conflict of interest.

C6. Please also provide an updated Competing Interests Statement declaring this commercial affiliation along with any other relevant declarations relating to employment, consultancy, patents, products in development, or marketed products, etc. 

Response: As indicated above, none of the co-authors claimed any conflict of interest. HMC is the principal public healthcare provider in the State of Qatar. It is a non-profit organization.

 

Reviewer 1

R1C1. I would only recommend authors to streamline introduction, that, as it is now, reads quite long.

Response: We refined the Introduction section in the revised manuscript to be more focusing.

---

## [Editor Report · Decision Letter 1]

3 Apr 2020

Harnessing Digital Health to Objectively Assess Cognitive Impairment in People undergoing Hemodialysis Process: The Impact of Cognitive Impairment on Mobility Performance Measured by Wearables

PONE-D-19-30554R1

Dear Dr. Bijan Najafi,

We are pleased to inform you that your manuscript has been judged scientifically suitable for publication and will be formally accepted for publication once it complies with all outstanding technical requirements.

With kind regards,

Luigi Lavorgna

Academic Editor

PLOS ONE
---

## [Editor Report · Acceptance letter]

8 Apr 2020

PONE-D-19-30554R1 

Harnessing Digital Health to Objectively Assess Cognitive Impairment in People undergoing Hemodialysis Process: The Impact of Cognitive Impairment on Mobility Performance Measured by Wearables 

Dear Dr. Najafi:

I am pleased to inform you that your manuscript has been deemed suitable for publication in PLOS ONE. Congratulations! Your manuscript is now with our production department. 

With kind regards,

on behalf of

Dr. Luigi Lavorgna 

Academic Editor

PLOS ONE